# PDQN - A Deep Reinforcement Learning Method for Planning with Long Delays: Optimization of Manufacturing Dispatching

## Abstract

Scheduling is an important component in Semiconductor Manufacturing systems, where decisions must be made as to how to prioritize the use of finite machine resources to complete operations on parts in a timely manner. Traditionally, Operations Research methods have been used for simple, less complex systems. However, due to the complexity of this scheduling problem, simple dispatching rules such as Critical Ratio, and First-In-First-Out, are often used in practice in the industry for these more complex factories. This paper proposes a novel method based on Deep Reinforcement Learning for developing dynamic scheduling policies through interaction with simulated stochastic manufacturing systems. We experiment with simulated systems based on a complex Western Digital semiconductor plant. Our method builds upon DeepMind's Deep Q-network, and predictron methods to create a novel algorithm, Predictron Deep Q-network, which utilizes a predictron model as a trained planning model to create training targets for a Deep Q-Network based policy. In recent years, Deep Reinforcement Learning methods have shown state of the art performance on sequential decision-making processes in complex games such as Go. Semiconductor manufacturing systems, however, provide significant additional challenges due to complex dynamics, stochastic transitions, and long time horizons with the associated delayed rewards. In addition, dynamic decision policies need to account for uncertainties such as machine downtimes. Experimental results demonstrate that, in our simulated environments, the Predictron Deep Q-network outperforms the Deep Q-network, Critical Ratio, and First-In-First-Out dispatching policies on the task of minimizing lateness of parts.

## 1 Introduction

This paper proposes a method based on Deep Reinforcement Learning for automated production scheduling in semiconductor manufacturing systems. In such systems, scheduling decisions must be made for processing operations. These systems involve machines that each perform operations to process a variety of different semiconductor devices. Each type of device requires a specific set of operations to be performed which varies depending on the type of device. These systems also include re-entrant connections, and the machines may be subject to machine failures, in which machines break down during processing and need to be repaired before resuming operations. In the industry, static dispatching policies such as Critical Ratio (CR) or First-In-First-Out (FIFO) are often used, together with manual adjustments at failing machines. Obtaining efficient scheduling and dispatch policies at every machine, especially in a system with machine failures, is a challenging and complex task. There are multiple key productivity indicators that are considered in manufacturing systems. Included in these are throughput, cycle time, and lateness with respect to due dates. Optimizing weighted combinations of these productivity indicators is a challenging task. In this paper we introduce a novel Reinforcement Learning algorithm for planning in such environments. Due to the complexity of the issue, in this paper we focus specifically on the task of reducing lateness with respect to due dates. It should be noted that this objective function, in itself, and in isolation, is a measure that manufacturing system researchers have studied for decades, with less than desired success. In each factory system modeled in this paper, there is a set of machines that are used to

process boxes of semiconductor devices on sheets of silicon wafers. These boxes are referred to as parts. At each point in time that a machine becomes available, a dynamic scheduling decision (dispatching decision) must be made as to what that machine should do next. In this paper, this decision is modeled as a choice of which part to process next from the queue at that machine group. Here, this process is modeled as a Markov Decision Process (MDP). An MDP includes a state, action, and reward where the state is the information describing the system, the action is a choice of which decision to make, and the reward is a signal returned after an action is taken which indicates the quality of that action. In the factory systems being modeled, the semiconductor devices produced are hard drive head chips which are used to read from and write to hard drives. There is a range of different head types produced in the facility. In our experiments, we simulate the first 20 processing steps for each head type to model the front of line in the facility.

In recent years Deep Reinforcement Learning (RL) techniques have been demonstrated to show remarkable performance in a number of previously challenging domains such as complex games (Mnih et al., 2015; Silver et al., 2016a;b; Berner et al., 2019; Vinyals et al., 2019; Schrittwieser et al., 2020). In addition, some work has been done in applying such techniques to manufacturing systems, e.g. Waschneck et al. (2018). One of the main difficulties in this domain is delayed rewards with particularly long delays; other domains with this feature include medical interventions in healthcare. Consequently, our focus in this paper is to develop a planning method that can account for extra long delays in rewards, together with highly stochastic dynamics.

So, this paper presents the Predictron Deep Q-Network (PDQN), a novel Deep RL technique that combines the Deep Q-Network (DQN) (Mnih et al., 2015) and predictron (Silver et al., 2017) methods to learn a policy for dispatching of parts in a simulated system modeled after a semiconductor manufacturing facility. The DQN is a model-free RL optimization algorithm that trains through experience to estimate Q-values which can be used to form a policy. The predictron is a model-based policy evaluation algorithm that can be rolled forward multiple "imagined" planning steps to predict future rewards and values. The PDQN uses the predictron as a trained background planning model to generate value estimates for use in fine-tuning a pre-trained DQN. By doing this, it is possible to incorporate background planning as part of the training process. This combination helps the algorithm account for the highly delayed rewards encountered in these factory systems. Inspiration for this setup comes in part from Dyna (Sutton, 1990) in which a model is used to train a policy, the main difference is that the PDQN uses an abstract model to perform its planning. The PDQN is also closely related to methods such as Value Prediction Network (VPN) (Oh et al., 2017) and MuZero (Schrittwieser et al., 2020), but is different from these papers and approaches by using background planning instead of decision time planning and by using an arbitrary number of steps in between each abstract state representation. Background planning has recently been shown to be the largest contributor to policy improvement when using planning in model-based RL (Hamrick et al., 2021). We compare the PDQN with both DQN and two standard factory dispatching policies, CR and FIFO.

## 2    RELATED WORK

### 2.1    DEEP REINFORCEMENT LEARNING IN PRODUCTION SCHEDULING

In recent years, work has been done in applying deep reinforcement learning to production scheduling tasks (Cadavid et al., 2019). Stricker et al. (2018) presents a Q-learning with artificial neural network function approximation dispatching method. This method was demonstrated to outperform a First-In-First-Out on the task of maximizing utilization and minimizing lead time on a small simulated semiconductor manufacturing system.

In Zhang et al. (2020) a method is similarly proposed to automatically generate priority dispatch rules using a deep reinforcement learning agent. Here, a Graph Neural Network-based scheme is used to embed the states. This work, however, only applies to job shop problems in which there are a fixed number of jobs to be completed. Therefore this approach would be inapplicable to the more realistic production scheduling problem encountered in semi-conductor manufacturing where new jobs are repeatedly being added into the system and production can continue indefinitely. In addition, the work in Zhang et al. (2020) only considers deterministic systems which don't account for uncertainties such as machine failures.

In Katsikopoulos & Engelbrecht (2003) an approach to addressing Markov decision processes with delays and asynchronous cost collection is considered. However that work assumes fixed or deterministic delays or stochastic delays which are independent of the state. In the systems considered in this paper the delays are dependent upon the policy so this approach may not be applicable. In Derman et al. (2021) they consider MDPs in which there are action delays such that actions are executed some number of steps after they are chosen. This is a separate issue than the one we are addressing with PDQN. In our case actions are executed immediately, the challenge in our case is that actions affect the return over many time steps. In Campbell et al. (2016) they consider the problem of applying Q-learning with stochastic time delays in the reward signal. This is again different from the issue we are addressing as in our problem. It is not that specific reward values are not received immediately it is that the effects of each action impact the return over ong horizons which makes learning and credit assignment difficult. Delayed feedback is also considered in Walsh et al. (2008). However it is limited to fixed, constant delays, which is not applicable for our case.

## 2.2 DEEP Q-NETWORK

DQN Mnih et al. (2015) is a Q-learning method, which works by estimating the expected discounted future return for a given state and action pair. For DQNs this is accomplished by using a Neural Network function approximator. DQN uses mini-batch stochastic gradient descent to update the weights of the neural network based on the gradient of a loss function to minimize the expected value of the loss. For DQNs the Mean Squared Error (MSE) loss is used as seen in Equation 1.

$$L(\theta) = \mathbb{E}[(r + \gamma \max_{a'} Q(s', a'; \bar{\theta}) - Q(s, a; \theta))^2]$$ (1)

where $\theta$ represents the weight parameters of the neural network. $\bar{\theta}$ represents an earlier copy of $\theta$, which is used to form a target network. The purpose of the target network is to estimate the expected value of the next state. The training is stabilized by only updating $\bar{\theta}$ after a number of training iterations, thereby giving the online network a stable target. An alternative to this is to use a soft update technique where the target is gradually updated towards the online network.

Q-learning is considered an off-policy method, as it can train on data collected by a different policy. To get the best result, however, exploration and exploitation should be tuned according to the problem. A common solution is to use the $\epsilon$-greedy policy on the Q-function, where $\epsilon$ is slowly decayed over time. Experienced data is stored as a set of {state, action, reward, next state} in an experience replay buffer and sampled according to some distribution, originally uniformly.

## 2.3 PREDICTRON

The predictron (Silver et al., 2017) is an architecture for model-based value estimation and policy evaluation. It consists of a fully abstract model which works by "imagining" a sequence of waypoints, each simultaneously describing an arbitrary number of steps into the future and an estimation of the value from the abstract waypoint state. It is strongly related to methods such as n-step TD-learning (Watkins, 1989) and eligibility traces (Sutton & Barto, 2018). The predictron learns a representation function $\mathbf{f}$ which outputs the first abstract state. Furthermore, it learns $K$ sets of functions, where each set includes a value function $\mathbf{v}^k$, a next abstract state function $\mathbf{s}^k$, a reward function for the transition to the next abstract state $\mathbf{r}^k$, a discount value function $\boldsymbol{\gamma}^k$ and an eligibility trace function $\boldsymbol{\lambda}^k$.

The predictron has two outputs describing the predicted returns (preturns), $\mathbf{g}^{0:K}$ and $\mathbf{g}^\lambda$. Here $\mathbf{g}^{0:K}$ is the set of $K$ preturns, with one k-preturn $\mathbf{g}^k$ for each abstract step $k$, as seen in Equation 2. The $\lambda$-preturn $\mathbf{g}^\lambda$, seen in Equation 3, is the weighted average of the k-preturns, where the $\lambda$-weights are determined using the learned eligibility trace parameters.

$$\mathbf{g}^k = \mathbf{r}^1 + \boldsymbol{\gamma}^1(\mathbf{r}^2 + \boldsymbol{\gamma}^2(\ldots + \boldsymbol{\gamma}^{k-1}(\mathbf{r}^k + \boldsymbol{\gamma}^k \mathbf{v}^k)\ldots))$$ (2)

$$\mathbf{g}^\lambda = \sum_{k=0}^{K} \boldsymbol{w}^k \mathbf{g}^k$$ (3)

where

$$\boldsymbol{w}^k = \begin{cases} (1 - \boldsymbol{\lambda}^k) \prod_{j=0}^{k-1} \boldsymbol{\lambda}^j & \text{if } k < K \\[2ex] \prod_{j=0}^{K-1} \boldsymbol{\lambda}^j & \text{otherwise} \end{cases} \tag{4}$$

The predictron is trained by minimizing the MSE loss for both $\mathbf{g}^{0:K}$ and $\mathbf{g}^\lambda$, as defined in Equations 5 and 6.

$$L^{0:K} = \frac{1}{2K} \sum_{k=0}^{K} \|\mathbb{E}_p[\mathbf{g}|s] - \mathbb{E}_m[\mathbf{g}^k|s]\|^2 \tag{5}$$

$$L^\lambda = \frac{1}{2} \|\mathbb{E}_p[\mathbf{g}|s] - \mathbb{E}_m[\mathbf{g}^\lambda|s]\|^2 \tag{6}$$

where $\mathbb{E}_p$ is the sampled sum of discounted rewards gained in the episode, and $\mathbb{E}_m$ is the predicted value from the model.

Thirdly, an optional loss is minimized, the consistency loss, improving the consistency between the k-preturns and the $\lambda$-preturn, as seen in Equation 7.

$$L = \frac{1}{2} \sum_{k=0}^{K} \|\mathbb{E}_m[\mathbf{g}^\lambda|s] - \mathbb{E}_m[\mathbf{g}^k|s]\|^2 \tag{7}$$

The predictron is the inspiration for both the VPN (Oh et al., 2017) and MuZero (Schrittwieser et al., 2020), in the sense that they all use abstract state-space representations. The main difference is that VPN and MuZero also train a policy for control, whereas the predictron is purely policy evaluation. VPN and MuZero both use tree searches to conduct their planning, whereas the predictron uses an estimation of eligibility traces to different depths of its abstract version of the expected future.

## 3 METHODS

This chapter describes the general setup of the methods used in this paper. Section 3.1 describes how the simulated factory is modeled as an MDP. Section 3.2 describes the CR and FIFO dispatching policies. Section 3.3 describes the proposed new PDQN method while the Neural Network architecture and hyperparameter setup for both the DQN and PDQN are described in the Appendix section A.1.

### 3.1 MDP MODELLING

#### 3.1.1 STATE SPACE REPRESENTATION

The state-space representation consists of the set of variables $S = S_{hs} \cup S_{hd} \cup S_{hp} \cup S_m$ representing the number of parts of each type at each sequence step, the number of parts due for each head type, the number of parts that are past due, and the machine that the dispatch decision corresponds to.

First, information about the work in process (WIP) is included in the state space. Let $N_{ht,j}$ be the number of parts in the factory of head type $ht$ at step $j$ of production. Included in the state space are the values of $N_{ht,j}$ for all combinations of head type $ht \in H$, where $H$ is the set of all head types and sequence step $j \in J_{ht}$, where $J_{ht}$ is the set of sequence steps of head type $ht$, $J_{ht} = \{1, 2, ..., M_{ht}\}$, where $M_{ht}$ is the number of sequence steps for head type $ht$. Let this set of state variables be $S_{hs}$, then $S_{hs} = \{N_{ht,j}|(ht \in H) \wedge (j \in J_{ht})\}$.

Second, information about the number of parts that are due for each head type is included in the state space. Let $D_{ht}$ be the number of parts due for head type $ht$. Let the set of due part state variables be $S_{hd}$, then $S_{hd} = \{D_{ht}|ht \in H\}$

Third, information about the parts which are past due are included as well. As such the values $P_{ht}$ representing the number of parts past due for each head type $ht$ are included as well. Let this set of state variables be $S_{hp}$, then $S_{hp} = \{P_{ht}|ht \in H\}$

Lastly, the state space includes the machine variables where $S_m$ is a one-hot vector indicating the machine that the dispatching decision for that time step corresponded to.

### 3.1.2 ACTION SPACE

An action is made every time a machine is ready to process a new part and a part is in its queue. The action determines which part to process next, and consists of selecting a head type and a sequence step. Let $H_m$ be the head types present in the queue for the affected machine $m$, then $A = (ht \in H_m, j \in J_{ht})$

### 3.1.3 REWARD SIGNAL

The reward signal is designed to penalize the agent for being late on parts with respect to their due dates. To accomplish this the reward at each time step is made to be negative and proportional to the amount of time that has elapsed during the time step and the number of parts that are past due. This can be represented as an integral over the time between two time steps. Let $t_i$ be the time at which time step $i$ occurs in the MDP representing the factory. Let $R_i$ be the reward for time step $i$.

$$R_i = -\int_{t_{i-1}}^{t_i} N_d(t)\,\mathrm{d}t \tag{8}$$

where $N_d(t)$ is the number of parts that are past due at time $t$. This way the agent will be repeatedly penalized each time step at a rate which is proportional to the number of parts that are past due. This serves as a heuristic that will encourage the agent to complete parts before their due dates.

### 3.2 CRITICAL RATIO AND FIRST-IN-FIRST-OUT DISPATCHING POLICIES

CR dispatching is a part priority rule based on the ratio between remaining time until due and the remaining time needed to complete processing of a part. The CR dispatching policy proceeds by selecting the part from the queue with the smallest critical ratio value. This will prioritize the parts which are most late and should therefore reduce the lateness of parts.

FIFO dispatching is done by selecting the part to process which was added to the queue first. When considering a single queue, FIFO may reduce max waiting time at that queue by always processing the part that has been waiting for the longest. However, this policy may not be optimal for the whole system as it does not take into account the state of other queues.

### 3.3 PDQN

Presented here is the PDQN algorithm. PDQN addresses the need for learning from highly delayed rewards and dynamic uncertainty due to machine downtimes by including abstract planning for a large number of steps. PDQN consists of two parts, an optimal decision policy determining component based on DQN, and an abstract planning trajectory-based value estimation component based on the predictron; this latter value estimate function of the predictron is fed to the DQN during training. In a traditional DQN, the model trains by minimizing the difference between its Q value estimate and a target value, as seen in Equation 1. This target is formed by sampling the reward for one step and then estimating the discounted return from the subsequent state using the same DQN architecture but with older weights, referred to as the target model. In the PDQN, we instead train the predictron part to estimate the value of states under the DQN policy, and use this value estimate as the target for the DQN, effectively substituting $\max_{a'} Q(s', a'; \bar{\theta})$ with $\mathbb{E}_m[\mathbf{g}^\lambda | s']$, as seen in Equation 9.

$$L(\theta) = \mathbb{E}[(r + \gamma \mathbb{E}_m[\mathbf{g}^\lambda | s'] - Q(s, a; \theta))^2] \tag{9}$$

where $\mathbb{E}_m[\mathbf{g}^\lambda | s']$ is the predictron estimate for the discounted return from the subsequent state $s'$ and $r$ is the reward given by the environment.

By using the predictron as target, the policy is trained using background planning. This is because the predictron part is trained to estimate the actual return from running the policy. By incorporating background planning, the policy can be trained towards better targets, which can take much more delayed rewards into account, and by using learned eligibility trace weights, it can learn to better

assign the right weight to late rewards. As the policy converges towards a better policy, the predictron will have to be updated to fit the policy again. Where, in traditional DQN, the target model is updated by simply copying the weights of the online model, we here update the target from the predictron by training it on new samples collected by the policy. This is done by fixing the policy for a number of steps and collecting samples of states along with the following $h$ rewards and the value of the h[th] state as estimated by the policy. It is important to note, that the predictron is trained in a supervised manner, where the data is collected using the policy. With $i$ as the time step, the target for the predictron $\mathbb{E}_p[\mathbf{g_i}|s_i]$ is defined as

$$\mathbb{E}_p[\mathbf{g_i}|s_i] = \sum_{n=0}^{h-1}(\gamma^n * R_{i+n}) + \gamma^h * max(Q(s_{i+h}, \mathbf{a}; \theta_i)) \tag{10}$$

where $R$ is the actual return from the following $h$ steps and $max(Q(s_{i+h}, \mathbf{a}; \theta_i))$ is the expected value of the h[th] state, estimated by the policy. Therefore the target for the predictron is biased by the estimate of the policy. When using a long horizon the policy will have a small effect on the total discounted return target, while the actual return from within the horizon will increase the variance of the target. Shorter horizon lengths, however, would be more biased by the policy and have lower variance. Consequently, there is a trade-off between these that may lead to different horizon lengths being optimal for different scenarios. The tested PDQN here uses a horizon higher than the estimated number of steps needed to complete on average more than two batches of parts at any given time of the environment. The loss for the predictron is then calculated using Equations 11 and 12 and the consistency update loss from Equation 7.

$$L^{0:K} = \frac{1}{2K}\sum_{k=0}^{K}\|\mathbb{E}_p[\mathbf{g_i}|s_i] - \mathbb{E}_m[\mathbf{g}^k|s]\|^2 \tag{11}$$

$$L^\lambda = \frac{1}{2}\|\mathbb{E}_p[\mathbf{g_i}|s_i] - \mathbb{E}_m[\mathbf{g}^\lambda|s]\|^2 \tag{12}$$

As a policy evaluation method, the expectation is that the predictron model can provide better estimates of long-term returns than the DQN based Q-value estimator itself. Better return estimates will then create stronger targets for learning when used to train the policy. We expect the target to be better as the predictron architecture can create an abstract planning model, including abstract states which are rolled forward to predict future returns weighed by a learned eligibility trace. The desirable features of the algorithm derive from the predictron and Eligibility Traces properties in incorporating: 1. The true delayed rewards, rather than inaccurate surrogates and 2. The most effective bias-variance trade-off, with the associated dimensional reduction. We hypothesize that these characterizes would enable it to firstly perform very well. Secondly, it would likely dominate stand-alone Q-learning approaches such as DQN and variants.

The choice of using the predictron for planning instead of using decision time planning methods, such as tree search, was based on the nature of the environment. Due to the large delays between when actions are taken and the completion of the parts those actions relate to, actions generally have very little effect on the expected return over the next few states. Therefore if a tree search should be used, it would need to be traversed to a high depth before seeing the outcome of the immediate action taken. By using the predictron with an external control policy, effectively using background planning, the policy can get an estimate of the long-term state value directly on the state. It is therefore the target for the policy that is trained using background training, and through this, the policy is indirectly trained through background training as well. The PDQN training algorithm is summarized in Algorithm 1 along with a detailed hyperparameter setup in the appendix section A.1. The predictron architecture used is shown in Figure 1. It is an alternation of the original architecture from Silver et al. (2017). The main change is that all convolutional layers are replaced with fully connected layers. This alternation is made as the state representation in our environment is not spatially related in the same manner as the environments used in the original paper, making local convolutional filters less meaningful.

## 4 EXPERIMENT

To evaluate the PDQN method, two different factory systems are used; a balanced factory system, which was a balanced version of a real factory system, and a randomly generated factory system,

Figure 1: The fully connected version of the original predictron implemented with 16 depth layers with individual weights. Each fully connected layer has 128 neurons, except for the output layers which have 1. $s$ is the input state while $s^{0:K}$ are the abstract states. $V^{0:K}$ are the estimated values for each abstract state. $r^{0:K-1}$ are the expected abstract rewards received for transitioning from one abstract state to the next. $\gamma^{0:K-1}$ is the expected abstract discount factors to apply for each abstract step. $\lambda^{0:K-1}$ is the expected eligibility assigned to each abstract step. As the abstract steps is arbitrarily long, $\gamma$ and $\lambda$ can vary as well.

which was made publicly available. We compared the mean lateness and the sum of lateness, which was the return received by the agent during an episode. We compared the performance against the CR, FIFO, and DQN policies[1]. For the evaluation, the DQN and PDQN policies were fixed, meaning that they were not allowed to train on the data from the test set.

## 4.1 ENVIRONMENT SETUP

Two environments are considered, a balanced 20 sequence steps (B20) factory system and a generated 20 sequence steps (G20) factory system.

The B20 factory setup was based on a subset of a real semiconductor manufacturing facility owned by Western Digital Corporation.

The G20 factory setup was based on additional data from Western Digital Corporation. However these sets were randomly generated in order to match the distributions of the data without including proprietary information. Details about factory settings are included in the appendix in Section A.2, and all the factory files needed to run the simulations for the generated systems are included in the supplemental materials.

## 4.2 TRAINING AND MODEL SELECTION

Initial experiments showed that when comparing the performance of the trained models of the same type, large variations were seen between the models, but a low variation was seen on the individual models. This was expected to come from the random initialization of the policies, as well as from the randomness used for exploration. To account for the variation seen between models, multiple models were trained for both DQN and PDQN. For each factory system, 10 different DQNs were trained. Each of these DQN models was then used to initialize a PDQN model training session. The PDQN models were trained for 5 iterations, each resulting in a separate model. The best models, in terms of lateness, were selected through validation simulations. Further training details, architecture, and hyperparameter setup are explained in detail in Section A.1.

## 4.3 MAIN TEST RESULTS

The best performing DQN and PDQN model from each environment was tested on 50 simulations with different sample paths. Each sample path was initialized with a seed to ensure a reproducible

---

[1]Initial experiments did not show an improved performance of the DQN by applying the extensions proposed in Hessel et al. (2018). Furthermore, MuZero was considered, as according to Hamrick et al. (2021) this method incorporates planning. However, the results from the implementation did not converge. This method was prohibitively computationally heavy to run on this complex environment. This type of online planning which incorporates MCTS may not be appropriate for such complex stochastic environments.

behavior from the environment when used to compare the performance of different policies. For comparison, the mean sum of lateness for the completed parts[2] is used.

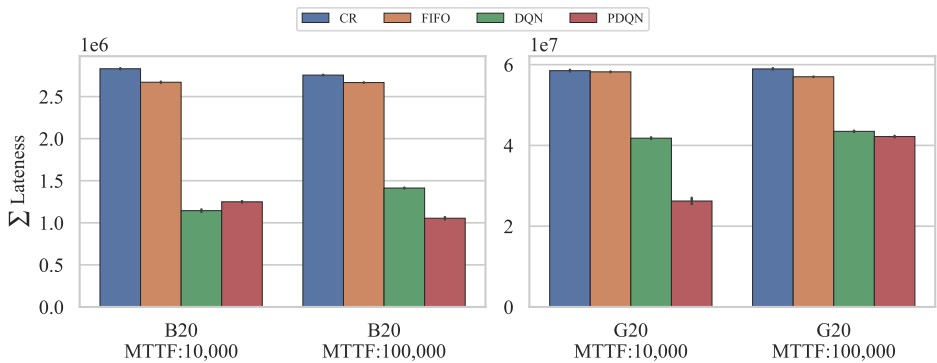

Figure 2: Here, the results from the objective function used to train the RL based policies are compared. The results are based on 50 test runs for each setup of each tested environment.

Figure 2 shows the results from the 50 test runs on all the tested environments. It is seen that for both factory systems, DQN and PDQN outperform CR and FIFO in terms of the mean sum of the lateness, which is the objective function used. Furthermore, PDQN outperform DQN on the same metric in all systems except the B20 system with MTTF of 10,000. The results seem to indicate that either DQN and PDQN perform similarly well, with very small difference in performance in the one case that DQN is superior, or the PDQN typically significantly outperforms DQN; we observe this in B20 with MTTF of 100,000 and G20 with MTTF of 10,000.

Overall, the results indicate that PDQN succeeds in improving the performance over the DQN policy, which again improves upon the standard factory dispatching policies, on the given lateness objective function in the tested environment. From a smart factory perspective, choosing PDQN with the correct parameters from the validation step appears to provide a robust close-to-optimal performance for the given objective function.

This performance improvement validates our initial hypothesis that incorporating the long delay reward data from the planning simulation, and the bias-variance features of the predictron, should together provide a powerful approach to addressing very long delays in highly dynamic, stochastic, and large-dimensional systems.

### 4.4 ADDITIONAL OBSERVATIONS TOWARDS FUTURE RESEARCH

In this sub-section, we provide some additional results to motivate a second objective function, integrating lateness and throughput. To be clear, we do not consider this objective function in this paper. An interesting observation is that for both DQN and PDQN, the sum of lateness often increases when the MTTF is increased for the two systems. By measuring the mean lateness and the number of completed parts from the 50 test runs, as seen in Figure 3, some additional insights to this increase, are made. The results can be seen in Tables 3 and 4 in the appendix section A.1.

PDQN outperforms the DQN in terms of mean lateness except in the B20 system with MTTF of 10,000. However, it is interesting to see how the number of completed parts, in general, is lower for the RL methods compared to the CR and FIFO policies. This result indicates that the used objective function of reducing the lateness of parts might result in unwanted behavior in terms of throughput, which is not part of the current objective function. Future work might need to consider an objective combining lateness and throughput. However, we do note that in the G20 system with an MTTF of 10,000, the PDQN throughput outperformed CR.

The G20 system has also been tested with WIP level of 15. The results from this test is shown in Table 4 in the appendix. The PDQN and DQN performance on those systems indicated that

---

[2]The initial 2× the WIP × the release batchsize number of parts are discarded from the results to only consider the steady-state system after a burn in period.

the learned policies favors higher WIP levels. This is possibly because higher WIP levels allow the policy to have greater flexibility, as more parts will line up at the machine queues, effectively increasing the number of allowed actions. It is interesting, however unexpected, that FIFO is better than CR on almost all parameters in the G20 setup, and also achieves lower mean lateness in the B20 setup. We suspect this result to be caused by the use of the CONWIP release policy and the high rate of machine failures. For example, CONWIP applied only to bottleneck machines would be more efficient. Future work should consider experimenting with a wide range of release policies.

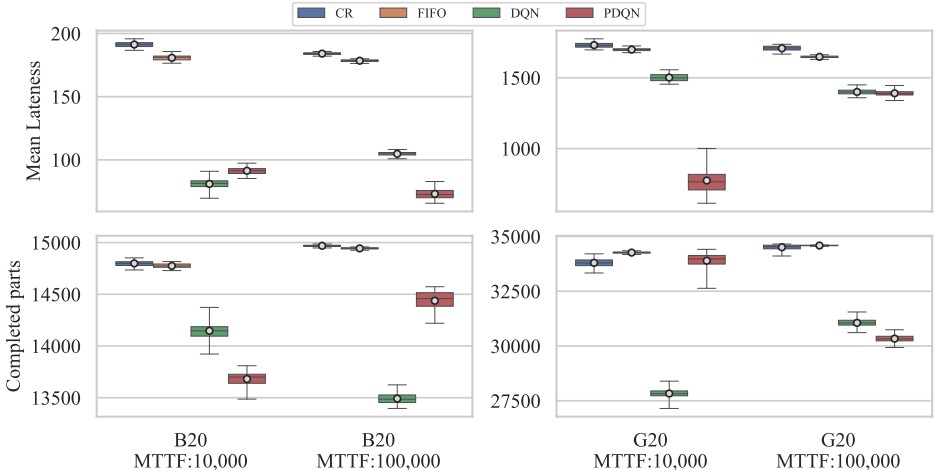

Figure 3: Additional observations regarding mean lateness (Top) and the number of completed parts (Bottom) are shown here. The data is shown with the inner quartiles (colored boxes), the median (black lines), the mean (white dots), and the min and max values (whiskers). Both the mean lateness and the number of completed parts is reported for all parts completed after the initial two full system runs, which are removed to align the observation with the objective function.

We observe that the validation step is fairly effective in the choice of hyperparameters including model parameters and iterations. Future research should experiment with hyperparameter optimization of the PDQN method to avoid this excess amount of training.

## 5 CONCLUSIONS

In this paper, we approach the problem of dispatching in simulated Semiconductor Manufacturing systems by using Deep RL techniques. We present the PDQN, a novel Deep RL approach combining DQN with value estimates from the predictron. We evaluate the use of both DQN and PDQN on two factory systems. The Deep RL methods are compared against CR and FIFO dispatching policies. The results show that PDQN outperforms CR, FIFO, and DQN in terms of lateness of part in the systems and that both Deep RL methods outperform CR and FIFO on this task.

From these results, we see that our hypothesis holds true, in that using the predictron architecture to better predict target values with very large delays, and provide powerful bias-variance trade-offs, can indeed increase the performance of a DQN based policy. That is, by incorporating this trained abstract planning model, the policy seems to better learn from delayed rewards in systems such as the dynamic manufacturing systems described here. Currently, however, there is a large variance in the performance of models after training, so carefully choosing a trained model by validating its performance is recommended.

Scheduling in semiconductor manufacturing facilities is a complex task that has a significant effect on the efficiency of production. Estimation of long-term values in this domain is especially difficult due to the nature of the factory systems. By including predictron methods, which have improved predictive ability, we were able to train models that outperform the DQN, CR, and FIFO on our chosen objective of reducing lateness of parts. The methods presented in this paper progress the application of Deep RL to scheduling algorithms in this domain.

REPRODUCIBILITY STATEMENT

All the code and the factory setting files are included in the supplementary material. To generate new factory systems use the fact_file_gen.py file, to reproduce the G20 system use seed=0. The PDQN algorithm is described in Algorithm 1.

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

## A APPENDIX

### A.1 ARCHITECTURE AND HYPERPARAMETER SETUP

Here we present the environment-dependent DQN and PDQN hyperparameter settings for the experiments. We used the same hyperparameter settings for all experiments. The hyperparameters were tuned based on experiments on the B20 system.

The architecture of the DQN was implemented as a 4 layer fully connected neural network, where the first three layers had 400, 250, and 125 neurons respectively, and used the ReLU activation function. The last layer was the output layer, which had $N_a$ number of neurons, where $N_a$ was the number of actions in the environment. The DQN was trained using the MSE loss with the Adam optimizer. The DQN was trained with a batch size of 32 and a learning rate of 0.005 using a discount factor $\gamma$ of 0.99. The target network was softly updated with $\tau = 0.125$. Actions were chosen using the $\epsilon$-greedy policy on the Q-function with $\epsilon$ starting at 1.0 and decaying with 0.999 at each step. A summary of the DQN hyperparameters is shown in Table 1.

The architecture of the predictron part, seen in Figure 1, is a fully connected version of the original predictron architecture (Silver et al., 2017), i.e., all convolution layers have been replaced with fully connected layers. All layers in the architecture, except for the output layers, have 128 neurons and use ReLU activations. The number of neurons for each abstract state-space representation serves as an encoding mechanism, where fewer neurons will result in a more compressed representation. Initial results show that using 128 neurons had a small positive effect on the performance when comparing to using 8, 16, 32, 64, and 256 neurons.

The depth of the predictron was set to 16, where each depth layer used its own weights. This configuration was chosen, as it was the best performing configuration in Silver et al. (2017). L2 regularization was used to counter overfitting. The training was conducted using the MSE loss with the Adam optimizer. The batch size of the predictron was set to 128, and the number of training batches per iteration was 128. The PDQN had the same discount rate as the DQN, $\gamma = 0.99$, and a horizon of 500, meaning that the target for the predictron was the sum of the actual discounted return of the following 500 steps and the discounted DQN value estimate for state $s_{i+500}$. With this setting we weighed the actual discounted return by 0.993, while weighing the remaining estimated value by 0.007.

$$\frac{\int_0^h \gamma^x dx}{\int_0^{\inf} \gamma^x dx} = \frac{\int_0^{500} 0.99^x dx}{\int_0^{\inf} 0.99^x dx} = 0.993$$

The horizon of 500 was found as 2.5 times the number of steps required to complete one batch on an empty system.

The policy part of the PDQN was initialized to be a copy of a pre-trained DQN. It was trained in the same manner as the DQN, with the only difference being that the target was estimated by the predictron part and that it was trained for 50,000 steps per iteration. A summary of the PDQN hyperparameters is shown in Table 2. The environment is set to run for an initial burn-in period using the greedy policy on the Q-function to skip the initial part of the environment before initiating the training.

First the predictron is trained for an initial number of steps, to allow it to learn a good estimate for the value of the policy. Then the policy is updated with the predictron value estimate as the target for an initial number of steps. Then the number of steps for training the predictron and the policy is set to a fixed value, and the alternation between training the two parts is continued for a number of iterations. During training, the Q function is updated after every step using the Adam optimizer on a small batch of data. The actions for the policy are sampled using epsilon greedy with a fixed epsilon. When training the predictron, the actions are sampled using the greedy policy on the Q function. The predictron is updated after all samples in that sequence have been collected in a supervised learning manner.

A.2 SETTINGS AND GENERATION OF FACTORY SYSTEMS

The balanced 20 sequence steps (B20) factory system was based on a subset of a real semiconductor manufacturing facility owned by Western Digital Corporation. In this system, the first 20 processing steps for each head type were simulated. The machine groups, process routings, and processing times in the simulated system were set to match the real system. Other factory parameters were set based on simulation results under the CR dispatch policy. Some machines were added to the simulated system to reduce bottle-necking at stations with high utilization. The system was tested with two machine failure rates to experiment with the robustness to changes in this parameter. The Mean Time To Failure (MTTF) was therefore set to either 100,000 minutes and 10,000 minutes, while the Mean Time To Repair (MTTR) was set to 120 minutes, and the WIP level was set to 30 release batches where each batch contained one part of each head type. The due date lead times and

Table 1: Hyperparameter setup for DQN

| Hyperparameter | Value |
|---|---|
| Learning rate | 0.005 |
| Batch size | 32 |
| Discount factor $\gamma$ | 0.99 |
| Exploration rate $\epsilon$ | $1.0 \rightarrow 0.02$ |
| $\epsilon$ decay rate | 0.999 |
| Soft update coefficient $\tau$ | 0.125 |
| Training steps | $500,000$ |
| Replay buffer size | $10,000$ |

Table 2: Hyperparameter setup for PDQN

| Predictron part | | Policy part | |
|---|---|---|---|
| Learning rate | 0.01 | Learning rate | 0.005 |
| L2 weight | 0.01 | Batch size | 32 |
| Batch size | 128 | Discount factor $\gamma$ | 0.99 |
| Batches per iteration | 128 | Exploration rate $\epsilon$ | 0.1 |
| Horizon $h$ | 500 | Steps per iteration | $50,000$ |
| Depth $k$ | 16 | Replay buffer size | $10,000$ |

WIP levels were set to ensure a mix of on-time and past due part completions. For each sample path in validation and testing, the environment was executed for 100,000 simulation minutes.

The generated 20 sequence steps (G20) factory systems were based upon data shared by Western Digital for a real factory system. These systems were designed to resemble real systems while being partially randomly generated to allow for sharing of the factory settings without divulging proprietary data. The G20 systems have the same (MTTF) and WIP levels as the B20 systems. The generation of the G20 system factory files is described in the appendix. In all setups, the release of parts into the system was controlled by a CONWIP based policy to maintain relatively constant levels of WIP in the system by releasing each batch of parts when another completes. Due dates were set using a set due date lead time for each head type, which specifies the time between when a part was released and when it was due. Machine failure and repair times were sampled from exponential distributions with specified mean times based on data from Western Digital.

Each G20 system included 10 head types. The process routings for each head type were sampled with replacement from a set of 24 stations. The processing times were sampled from a gamma distribution which was set to match the distribution of the balanced factory system. The number of machines in each station was selected to match the level of demand at each station. The MTTF was again set to 100,000 minutes and 10,000 minutes with an MTTR of 102 minutes. The level of WIP was set to 30 release batches, where each released batch contained one part of each head type. For each sample path in validation and testing, the environment was executed for 500,000 simulation minutes. The longer execution times for these setups were set to account for a lower throughput compared to the B20 system.

In all setups, the release of parts into the system was controlled by a CONWIP based policy to maintain relatively constant levels of WIP in the system by releasing each batch of parts when another completes. Due dates were set using a set due date lead time for each head type, which specifies the time between when a part was released and when it was due. Machine failure and repair times were sampled from exponential distributions with specified mean times based on data from Western Digital.

---

**Algorithm 1:** PDQN training

---

Q = Load pretrained DQN;
Start and run environment for an initial burn-in period, using the greedy policy on the Q
  function;
$predictron\_train\_steps = predictron\_batch\_size * batches\_per\_iteration$;
$TrainPolicy = False$;
$i = 0$;
**while** *training* **do**
    Update state: $\hat{s} = \hat{s}'$;
    Update allowed actions: $\hat{c} = \hat{c}'$;
    Use greedy policy on Q function to find action: $\hat{a} = argmax(Q(\hat{s}, \epsilon)|\hat{c})$;
    Take step: $\hat{s}', \hat{r}, \hat{c}' = step(\hat{a})$;
    Save to replay buffer: $ReplayBuffer.add(\hat{s}, \hat{a}, \hat{r}, \hat{s}', \hat{c}')$;
    Increase step counter: $i+ = 1$;
    **if** $TrainPolicy$ **then**
        Sample batch: $s, a, r, s', c' = sample_{batch}(ReplayBuffer)$;
        Calculate loss from Equation 9;
        Use Adam optimizer on batch loss;
        **if** $i >= policy\_steps\_per\_iteration$ **then**
            $TrainPolicy = False$;
            $i = 0$;
        **end**
    **else**
        $S_i = \hat{s}$;
        $R_i = \hat{r}$;
        **if** $step >= h$ **then**
            $\mathbb{E}_p[\mathbf{g}|S_{i-h}] = \sum_{n=0}^{h}(\gamma^n * R_{i-h+n}) + \gamma^h * max(Q(\hat{s}')|\hat{c}')$;
            Append $S_{i-h}, \mathbb{E}_p[\mathbf{g}|S_{i-h}]$ to $train\_data$;
            **if** $step >= predictron\_train\_steps$ **then**
                **while** $train\_data$ *is not empty* **do**
                    $s, \mathbb{E}_p[\mathbf{g}|s] = get_{batch}(train\_data_{predictron})$;
                    Calculate k loss from Equation 5;
                    Calculate $\lambda$ loss from Equation 6;
                    Use Adam optimizer on k loss and $\lambda$ loss;
                    To use consistency updates do:
                    Calculate cu_loss from Equation 7;
                    Use Adam optimizer on cu_loss;
                **end**
                $TrainPolicy = True$;
                $i = 0$;
            **end**
        **end**
    **end**
**end**

---

### A.3 ADDITIONAL OBSERVATIONS

Here, more results from running the dispatching methods on the two factory systems introduced in section 4, are presented. The mean lateness and the number of completed parts are reported for the entire length of the simulation, except for the initial $2\times$ WIP $\times$ the release batchsize number of parts, from which the results are discarded. The general trend is that the RL-based methods outperform CR and FIFO in terms of mean lateness, where PDQN outperforms DQN in 4 of 6 setups. CR and FIFO have the highest number of parts completed in the 15 and 30 WIP level experiments. Note that the number of completed parts is low while the mean lateness is also low for both the DQN and PDQN methods. The reason for this might be that the objective function used penalizes the agent from completing parts later than their due time. As the CONWIP release policy is used, the agents

might learn to delay the completion of batches as much as possible to delay the introduction of new parts for as long as possible. This way, the mean lateness can be low if the majority of parts get completed fast, and the number of completed parts will be low, due to the delayed completion of the last part of the batch, which would match the results seen.

Table 3: Test results for balanced 20 step factory, B20, on 100,000 simulation minutes. The results are averaged over 50 different test runs and shown with $\pm$ standard deviation of the sample mean over the different test runs.

| | CR | FIFO | DQN | PDQN |
|---|---|---|---|---|
| **30 WIP BATCHES** | | | | |
| **MTTF** | | **10,000** | | |
| Sum of Lateness | 2.83e6±3.92e3 | 2.67e6±3.68e3 | **1.14e6±8.00e3** | 1.25e6±4.61e3 |
| Mean Lateness | 191.3±0.30 | 180.7±0.28 | **80.9±0.62** | 91.3±0.39 |
| Completed parts | **14798.7±3.90** | 14774.2±3.18 | 14146.8±12.52 | 13680.6±10.78 |
| **MTTF** | | **100,000** | | |
| Sum of Lateness | 2.76e6±1.97e3 | 2.67e6±1.74e3 | 1.41e6±2.68e3 | **1.05e6±7.92e3** |
| Mean Lateness | 184.1±0.13 | 178.4±0.12 | 104.7±0.28 | **73.1±0.39** |
| Completed parts | **14968.9±1.26** | 14943.7±1.12 | 13491.3±7.16 | 14437.5±12.97 |

Table 4: Test results for generated 20 step factory, G20, on 500,000 simulation minutes. The results are averaged over 50 different test runs and shown with $\pm$ standard deviation of the sample mean over the different test runs. 2 different failure rates and 2 different WIP levels are compared.

| | CR | FIFO | DQN | PDQN |
|---|---|---|---|---|
| **15 WIP BATCHES** | | | | |
| **MTTF** | | **10,000** | | |
| Sum of Lateness | 19.3e6±27.1e3 | 18.4e6±16.5e3 | **13.2e6±24.8e3** | 14.9e6±24.2e3 |
| Mean Lateness | 561.4±0.74 | 533.4±0.47 | 489.9±1.29 | **464.6±0.96** |
| Completed parts | 34417.2±9.20 | **34540.5±7.01** | 26919.6±28.99 | 32060.0±38.26 |
| **MTTF** | | **100,000** | | |
| Sum of Lateness | 18.6e6±16.7e3 | 17.9e6±10.5e3 | 13.4e6±22.5e3 | **13.0e6±20.4e3** |
| Mean Lateness | 532.7±0.45 | 512.4±0.30 | **428.5±0.68** | 449.3±0.82 |
| Completed parts | **34926.1±3.72** | 34902.6±2.07 | 31311.7±18.01 | 28828.8±24.01 |
| **30 WIP BATCHES** | | | | |
| **MTTF** | | **10,000** | | |
| Sum of Lateness | 58.5e6±115.2e3 | 58.2e6±48.0e3 | 41.8e6±99.2e3 | **26.2e6±399.2e3** |
| Mean Lateness | 1730.8±2.55 | 1698.9±1.45 | 1501.6±3.68 | **774.6±12.65** |
| Completed parts | 33798.6±25.51 | **34265.2±5.32** | 27831.6±27.91 | 33895.5±44.15 |
| **MTTF** | | **100,000** | | |
| Sum of Lateness | 58.9e6±102.5e3 | 57.0e6±35.8e3 | 43.5e6±87.7e3 | **42.2e6±94.4e3** |
| Mean Lateness | 1707.9±2.45 | 1647.9±1.05 | 1400.2±2.64 | **1390.7±2.95** |
| Completed parts | 34508.6±16.75 | **34587.8±2.22** | 31057.7±26.64 | 30336.1±25.41 |

