# OpenReview forum: "PDQN - A Deep Reinforcement Learning Method for Planning with Long Delays: Optimization of Manufacturing Dispatching"
_ICLR.cc/2022/Conference — ICLR 2022 Submitted_

### Official Review · Reviewer_8ES3 · 2021-10-29

**Correctness:** 2
**Technical Novelty And Significance:** 1
**Empirical Novelty And Significance:** 1
**Recommendation:** 3
**Confidence:** 4

**Main Review:**

*Main Review

Pros:
1. The problem is very interesting. And it is a real life application that could generate good real-life values.
2. The code is released and has good reproducibility.

Cons:
1. It is not clear why the authors decide to make the presented modifications to the original predictron algorithm.
And it is hard to understand the intuition of the algorithm.
The modifications are not justified.

2. There’s no proper reinforcement learning baselines.
The algorithm is only compared with some toy methods and the RL baselines DQN is quite outdated. If I didn’t misunderstand, at least predictron should be compared against?
And there are also some new algorithms that seem to be applicable to the problem.



**Summary Of The Paper:**


 PDQN - A Deep Reinforcement Learning Method for Planning with Long Delays: Optimization of Manufacturing Dispatching
In this paper, the authors applied a reinforcement learning algorithm to the problem of manufacturing dispatching.
They used a model-based algorithm that is inspired on the algorithm of predictron.


**Summary Of The Review:**


*Summary Of The Review
I am worried about the novelty of the algorithm, and the claims are not supported by experiments.

---

> ### Author Response · Authors · 2021-11-20
> **Response to review**
>
> We appreciate the reviewer's review and feedback.
>
> Unless we misunderstand something, we believe there has been some significant misunderstanding of the predictron method used for planning as part of the PDQN.
>
> Below are our responses to the reviewer's concerns and questions, where we also try to briefly provide an explanation of the properties of the predictron. In addition, please review our explanation at a meta- review level above.
>
> Reviewer:
>
> It is not clear why the authors decide to make the presented modifications to the original predictron algorithm. And it is hard to understand the intuition of the algorithm. The modifications are not justified.
>
> Author response:
> - DQN and its variants are focused on getting Q-learning to work when the Q-function is parametrized by a deep learning network (-1. Experience replay including use of mini batches 2. Target network creation for stable parameter updates). However, it is important to note that the targets range from TD(0) and MC to combinations.
> - By way of contrast, we use the Predictron as a target, to create a target which can achieve enhanced bias-variance trade-offs, by combining different episode lengths effectively.Use of the Predictron achieves this, going beyond forward-looking eligibility traces (since we have the full horizon bound episodes) by incorporating value-function based representations. Note that the predictron is trained using supervised learning on batches of h sequential steps collected using the epsilon greedy policy on the Q-network.
> - A key observation is that the Predictron architecture incorporates planning via the depth feature. This in turn implies that the Predictron-based target incorporates implicit planning, while most other approaches either lack this planning feature, or have much cruder planning, or have very computationally intense planning.
> - Next, observe that the PDQN is a combination of the two methods DQN and Predictron. However, the Predictron is not by itself able to make decisions as it tackles the Markov Reward problem, and DQN is not by itself able to account for eligibility traces and planning in general.
> - The PDQN, therefore, uses the strong policy evaluation ability gained from using the predictron as target for the DQN method. Thereby introducing background planning to the DQN.
>
> Reviewer:
>
> There are no clear, existing reinforcement learning baselines. The algorithm is only compared with some toy methods and the RL baselines DQN is quite outdated. If I didn’t misunderstand, at least predictron should be compared against? And there are also some new algorithms that seem to be applicable to the problem.
>
> Author response:
> - The predictron can not be compared against as it does not produce policies for selecting actions, but rather only evaluate existing policies.
> - The FIFO and CR methods compared against are commonly used in the industry for complex systems, so we consider them to be relevant benchmarks.
> - Other planning reinforcement learning approaches have been considered, however, they are often either
>   - Computationally heavy, leading to lack of convergence
>      - Example: We considered MuZero, as it is state-of-the-art real-time planning and (in some sense) background planning, and dominates Value Prediction Network. However, as explained in the paper, MuZero did not converge for our problem.
>   - Only consider a small number of steps into the future
>      - We considered N-step methods for DQN, and other extensions, however, they did not seem to improve the performance of the DQN, which is why they are not used as baselines.

---

### Official Review · Reviewer_uJaW · 2021-10-30

**Correctness:** 4
**Technical Novelty And Significance:** 2
**Empirical Novelty And Significance:** 3
**Recommendation:** 5
**Confidence:** 4

**Main Review:**

Strengths:
- The authors develop a realistic, public shareable environment for manufacturing dispatching that is suitable for RL methods research.
- The PDQN framework is novel and offers better performance for these tasks.

Weaknesses:
- The main baseline for the work is quite old—the authors use the 2015 vanilla DQN. The authors say they tried various improvements from Hessel et al., and they didn't seem to help, but they don't enumerate which improvements they tried (see question 1 below). From a methods perspective, I find the evaluation of the PDQN to not be rigorous enough—the authors only evaluate in the new environment they constructed and they only compare against vanilla DQN.
- From an application perspective, the issue of sim2real problems does not receive _any_ attention that I can see. The authors do not go into detail about how they build their simulated environment—specifically how they choose incoming parts that need to be processed. They also do not discuss a real-world test.
- The throughput issue seems like a substantial real-world problem. I can't imagine wanting to implement this system if throughput (which translates directly to the bottom line) could decrease by 15%.

Questions:
1. The architecture the authors propose seems quite similar to a dueling DQN in the way that the Q-function estimation and the value function estimation are split—the difference being that the dueling DQN does not use a predictron and that the dueling DQN has a different aggregation architecture. The authors mention they tried the improvements of Hessel et al., but did they try the dueling architecture specifically? It would also be easy to integrate a predictron into the value-estimation network in a dueling DQN. (It would be interesting to know how much the predictron structure is specifically contributing.)
2. The use of L2 regularization in the predictron is interesting—did the authors find regularization improved performance at convergence or did it decrease sample complexity? There has been recent interest in the use of L2 regularization, e.g., https://arxiv.org/abs/1810.00123

Detailed comments:
- Eq. 1 is often called the Mean Squared Bellman Error because it measures the extent to which Q satisfies the Bellman equations
- “Q-learning is considered an off-policy method” -> “Q-learning is an off-policy method”. It’s definitional—doing Q-learning would not make - sense because there would be no exploration.
- Figure 1 is many pages after it is referenced

Post-rebuttal: It seems that the rebuttal discussion has been productive—the paper has been improved in many relevant ways. There are still outstanding issues—it seems like throughput and baselines are the largest of these.


**Summary Of The Paper:**

The authors study a deep reinforcement learning approach to the problem of scheduling machines in a semiconductor factory. The authors' main contribution is a deep Q-learning based system that uses a predictron to estimate the value function (instead of max_a Q(s',a)). The intuition behind the domain is that there are many machine groups that can process parts. The goal is to select the order that the machines should process parts. The authors compare a double DQN to their predictron DQN (PDQN) to baselines that are currently in use: critical ratio and first-in-first-out, with the objective of minimizing lateness. They build simulators based on real data: one that is proprietary and one that they release. They find the RL systems reduce lateness by substantially (20-50%) relative to the baselines and that their PDQN performs better by around 30% compared to DQN. However, they note that the RL systems complete substantially fewer parts, as much as 15% fewer for PDQN and 25% for DQN, suggesting that lateness may not be a good reward function for this task.

**Summary Of The Review:**

From a methodological perspective, I found the evaluation to not be rigorous enough. As an application perspective, there are critical application-specific issues that are not addressed.

---

> ### Author Response · Authors · 2021-11-20
> **Response to review**
>
> We appreciate the reviewer's thorough review and constructive feedback.
>
> Below are our responses to the reviewer's concerns and questions.
>
> Reviewer:
>
> The main baseline for the work is quite old—the authors use the 2015 vanilla DQN. The authors say they tried various improvements from Hessel et al., and they didn't seem to help, but they don't enumerate which improvements they tried (see question 1 below). From a methods perspective, I find the evaluation of the PDQN to not be rigorous enough—the authors only evaluate in the new environment they constructed and they only compare against vanilla DQN.
>
> Author response:
> - The goal of this paper was to develop methods specifically for this domain as this problem is of particular interest to the industry. Therefore we focused on results in these manufacturing settings.
> - We chose to compare against vanilla DQN as the PDQN is intended to build upon the vanilla DQN specifically.
> - The extensions to the DQN we compared with (although we did not provide detailed results in the submission) include N-step DQN, Double Dueling DQN, Prioritized Experience Replay, and the full Rainbow setup.
> - We have mentioned this in the text as well so that readers are aware of a state-of-the-art comparison.
>
> Reviewer:
>
> From an application perspective, the issue of sim2real problems does not receive any attention that I can see. The authors do not go into detail about how they build their simulated environment—specifically how they choose incoming parts that need to be processed. They also do not discuss a real-world test.
>
> Author response:
> - We agree, moving from simulation to the real system has not been described. This is an issue that is relevant to many RL applications. Here we focused on validating this new method in simulated systems so we consider sim2real to be beyond the scope of what we sought to describe in this paper.
> - The incoming parts were chosen using a batched CONWIP method, additional information has been added to the appendix describing how this method controls how incoming parts are released into the system.
>
> Reviewer:
>
> The throughput issue seems like a substantial real-world problem. I can't imagine wanting to implement this system if throughput (which translates directly to the bottom line) could decrease by 15%.
>
> Author response:
> - We agree and are working on making the throughput part of the reward from the environment in future work.
> - However, that does detract from the current results, which show that PDQN dominates in optimizing another key performance measure in manufacturing systems, namely the objective function of minimizing lateness which we focus on in this paper.
>
> Reviewer:
>
> The architecture the authors propose seems quite similar to a dueling DQN in the way that the Q-function estimation and the value function estimation are split—the difference being that the dueling DQN does not use a predictron and that the dueling DQN has a different aggregation architecture. The authors mention they tried the improvements of Hessel et al., but did they try the dueling architecture specifically? It would also be easy to integrate a predictron into the value-estimation network in a dueling DQN. (It would be interesting to know how much the predictron structure is specifically contributing.)
>
> Author response:
> - A main difference between the Dueling DQN and the PDQN is that the PDQN uses the value from the predictron as the target for the DQN, not as part of the actual DQN architecture.
> - The dueling DQN uses the sum of the separate value and advantage functions for outputting state/action values.
> - We did try to use the predictron as the Value branch for the dueling architecture, which has several advantages, such as running optimization and end-to-end training. This is an ongoing part of our research but it requires further work to refine and validate this method.
>
> Reviewer:
>
> The use of L2 regularization in the predictron is interesting—did the authors find regularization improved performance at convergence or did it decrease sample complexity? There has been recent interest in the use of L2 regularization, e.g., https://arxiv.org/abs/1810.00123
>
> Author response:
> - L2 regularization was mainly used to utilize the high number of neurons (compared to the expected minimum state-space representation) in the abstract state-space representation throughout the Predictron architecture. This was chosen instead of making a hyperparameter search for the number of neurons used.
> - The performance seemed to be comparable with and without L2 regularization.

---

### Official Review · Reviewer_QqV1 · 2021-11-02

**Correctness:** 3
**Technical Novelty And Significance:** 2
**Empirical Novelty And Significance:** 2
**Recommendation:** 3
**Confidence:** 4

**Main Review:**

Strengths
- The paper tackles an important decision/scheduling problem in manufacturing industry, which seems to be a very promising real-world application of deep RL.
- The proposed method outperformed the common heuristics used in industry in terms of cumulative lateness.
- The use of model-based approach to address the long reward delay issue in this problem makes sense to me.


Weaknesses and other comments
- The paper should formally define the dynamic scheduling problem in manufacturing in mathematical terms. The current level in the introduction is not sufficient for a scientific paper. The true objective of interest should also be clearly defined. The current choice of reward is the total time past due over the parts during a unit time period, but later in the experiments, it appears that the throughput is also a metric of interest. The choice of reward should be motivated by the ultimate objective of the problem, not by certain behavior of the policy that we want to encourage.
- Before the conclusion, it is mentioned that "It is interesting, however unexpected, that FIFO is better than CR on almost all parameters in the G20 setup, and also achieves lower mean lateness in the B20 setup." Is CR typically expected to outperform FIFO? Does this observation mean that the simulator requires some calibration before being used for training the agent?
- The baselines for the benchmarking should be more properly selected. For example, Section 2.1 discussed several recent works based on RL for the same problem. Why is not one of them selected as a baseline? The proposed method is basically DQN with the targets estimated by predictron based on rolling out a simulator. To justify this combination of techniques (especially the use of predictron), benchmarking should be performed against a version of DQN, say, with targeted estimated simply by MC rollouts.

**Summary Of The Paper:**

This paper tackles the dynamic scheduling problem in semiconductor manufacturing using an RL approach. Due to the processing actions taking different amount of time to complete, the problem has a long reward delay issue, which the paper addresses through using predictron to estimate the targets in DQN. Simulation experiments were performed on domains constructed using real-world data and against common heuristic baselines and DQN. Results show that the proposed method outperforms the baselines in terms of cumulative lateness of the parts.

**Summary Of The Review:**

This paper tackles an interesting and promising industrial application of RL. To address the long reward delay issue, a combination of DQN and predictron (for estimating targets) is proposed to train the agent, which makes sense. Empirical results are presented to show the advantage of the proposed method over common heuristics and vanilla DQN. The paper needs significant improvement in formal problem/objective definitions and the choice of baselines in benchmarking for a more convincing case. The simulation environment is not a standard one, so some calibration results should be presented to ensure the validity of the results.

---

> ### Author Response · Authors · 2021-11-20
> **Response to review**
>
> We appreciate the reviewer's thorough review and constructive feedback.
>
> Below are our responses to the reviewer's concerns and questions.
>
> Reviewer:
>
> The paper should formally define the dynamic scheduling problem in manufacturing in mathematical terms. The current level in the introduction is not sufficient for a scientific paper. The true objective of interest should also be clearly defined. The current choice of reward is the total time past due over the parts during a unit time period, but later in the experiments, it appears that the throughput is also a metric of interest. The choice of reward should be motivated by the ultimate objective of the problem, not by certain behavior of the policy that we want to encourage.
>
> Author response:
> - We agree and are working on making the throughput part of the reward from the environment.
> - However, in this paper, our objective function was lateness. Our intention was merely to show that RL methods can perform well or better than current methods for this objective function and that PDQN is a new planning approach for improving upon DQN in such highly delayed environments, for the reasons that we provide. Throughput was not part of our objective function. Addressing a combined delay-throughput objective function was/is part of our future research.
> - Since we are introducing a new algorithm, we decided to limit the scope of the problem we are addressing. This has been better clarified in the introduction to the paper.
>
> Reviewer:
>
> Before the conclusion, it is mentioned that "It is interesting, however unexpected, that FIFO is better than CR on almost all parameters in the G20 setup, and also achieves lower mean lateness in the B20 setup." Is CR typically expected to outperform FIFO? Does this observation mean that the simulator requires some calibration before being used for training the agent?
>
> Author response:
> - We are curious as to why we see this phenomenon and expect that it might be caused by the CONWIP policy used to release parts into the system. We do not believe that the issue is connected with calibrating the simulator.
>
> Reviewer:
>
> The baselines for the benchmarking should be more properly selected. For example, Section 2.1 discussed several recent works based on RL for the same problem. Why is not one of them selected as a baseline? The proposed method is basically DQN with the targets estimated by predictron based on rolling out a simulator. To justify this combination of techniques (especially the use of predictron), benchmarking should be performed against a version of DQN, say, with targeted estimated simply by MC rollouts.
>
> Author response:
> - Section 2 has been expanded with additional related works as well as reasons why they are not appropriate in this context.
> - Sampling rewards for a larger number of steps as is done for predictron training targets could be possible. Higher step returns will increase variance which may make training more difficult for the DQN. The predictron is better able to train on these high variance targets, due to the improved bias-variance trade-off resulting from its architecture. In preliminary tests, training the DQN with larger numbers of steps did not improve performance so we did not include such policies in our benchmarks.

---

### Official Review · Reviewer_ZLM1 · 2021-11-03

**Correctness:** 3
**Technical Novelty And Significance:** 2
**Empirical Novelty And Significance:** 2
**Recommendation:** 5
**Confidence:** 4

**Main Review:**

*Strengths*

- This work raises an important issue that has hardly been tackled in the RL literature, namely high delays in the reward signal. It has a strong motivation, and the proposed solution is sound in my opinion.
- In terms of contribution, I think that making a connection between a typical operations research problem and a deep RL method constitutes a contribution in and of itself, as I am not aware of any previous work that does so.
- The paper is clearly written, in a comprehensive manner that makes it easy to follow, may it be for an RL expert reader or for a scheduling one. I have minor comments in that regard, which I will make at the end of this review.

*Weaknesses / Questions to the authors*

- The authors cite Stricker et al. (2018) as an RL paper tackling scheduling tasks with Q-learning. Why is their method not used as a baseline in that paper? Indeed, PDQN is also a Q-learning algorithm and aims to solve the same type of problem. Another Q-learning method for delayed rewards is proposed in [3]...
- The authors motivate the predictron submodule by « highly delayed rewards » (Sec. 3.3 and a few other places). How much is it « high »? Although I guess it is stochastic, is it bounded? Another concern is regarding the way time steps are set in the MDP: according to Eq. (8) time is discretized in order for the reward signal to fit with the MDP framework. But what is a typical value of $\Delta_i:= t_i - t_{i-1}$? Would not it affect that of the reward delay, i.e. high $\Delta_i$ implies lower delay value? Similarly, what is a typical value for $h$ in Sec. 3.3.? I guess this should depend on an average delay value. How would a high $h$ value affect the computational time of PDQN (higher $h$ implies long-term inference)?
- How different are Eq. (11) and (12) from Eq. (5) and (6), respectively? If these are in some sense equivalent, the authors can synthesize their notation and refer to (5) and (6) in Sec. 3.3. instead of rewriting them.
- Sec. 2 looks like a Preliminary section providing the problem setting, more than a Related work section. In that regard, a review of delayed RL works is missing. More precisely, high delay in the signal results in some challenging issues: although one could perform embedding and retrieve a non-delayed MDP as in [1], this greatly increases computational complexity. In my opinion, the predictron has a major advantage over augmentation methods that may be even less tractable than MCTS (such drawback of MCTS is indeed mentioned in Sec. 3.3).
- In [2], the authors use a forward model to tackle execution delay issues. How does the predictron compare to this type of model? The use of eligibility traces seems to have similar effects.
-  In Figs. 2-3, DQN seems to perform decently against PDQN. This discredits a bit the use of the predictron. How could we explain such good performance of DQN which, by the way, does not account for delay at all? Unexpectedly, although they are fashioned precisely to solve such scheduling problems, FIFO and CR perform much worse than those two deep RL methods. How can this be interpreted?

**Summary Of The Paper:**

This paper finds its motivation in scheduling for semiconductor manufacturing systems. It proposes a deep RL algorithm for tackling the highly delayed feedback usually encountered in this type of dynamic system. The authors introduce an algorithm called Predictron Deep Q-Network (PDQN) that plugs a predictron architecture into the loss of the Q-network. The efficiency of PDQN is tested on two scheduling problems and challenged against 3 natural baselines, CR, FIFO, and DQN, the first two baselines being standard for scheduling.

**Summary Of The Review:**

Although this work is well-motivated, I have two main concerns leading me to a « weak reject » notation.
- First, on the theoretical side, the problem of delayed reward is not tackled frontally, but rather implicitly through the predictron. As such, delay is not formalized, although intrinsic to the problem considered. It weakens this paper’s contribution in my opinion, as one could question whether PDQN would necessarily work in a different type of domain (e.g., healthcare, as mentioned in Sec. 1), where the reward function has other dynamics.
- Second, delayed reward and delay in general (delayed execution; delayed observations), have been addressed in RL. This span of relevant works is never mentioned nor discussed. I list below a few references that are welcome to be discussed/compared with PDQN.

*Relevant references (all of them are missing in this paper)*

[1] Konstantinos V Katsikopoulos and Sascha E Engelbrecht. Markov decision processes with delays
and asynchronous cost collection. IEEE transactions on automatic control, 48(4):568–574, 2003.

[2] Derman, Esther, Gal Dalal, and Shie Mannor. "Acting in Delayed Environments with Non-Stationary Markov Policies." ICLR 2021.

[3] Jeffrey S Campbell, Sidney N Givigi, and Howard M Schwartz. Multiple model q-learning for stochastic asynchronous rewards. Journal of Intelligent & Robotic Systems, 81(3-4):407–422, 2016.

[4] Thomas J Walsh, Ali Nouri, Lihong Li, and Michael L Littman. Learning and planning in environments with delayed feedback. Autonomous Agents and Multi-Agent Systems, 18(1):83, 2009.

---

> ### Author Response · Authors · 2021-11-20
> **Response to review (1/2)**
>
> We appreciate the reviewer’s thorough review and insight into the paper, as well as the constructive feedback.
> We also appreciate that the reviewer has explicitly remarked on the strong motivation for the domain, as well as the soundness of the work.
>
> Below are our responses to the reviewer’s concerns and questions.
>
> Reviewer:
>
> The authors cite Stricker et al. (2018) as an RL paper tackling scheduling tasks with Q-learning. Why is their method not used as a baseline in that paper? Indeed, PDQN is also a Q-learning algorithm and aims to solve the same type of problem. Another Q-learning method for delayed rewards is proposed in [3]...
>
> Author response:
> - Stricker et al. (2018) address a slightly different task. They use a Q-learning method with eligibility traces and ANN function approximator to solve the problem of moving a lift around in the factory while dispatching. The method is probably interesting, however, it cannot use replay buffers, as it is using eligibility traces in a backward view manner. Furthermore, the application is in the same area but applied in a different way. Furthermore, the Predictron uses eligibility traces more effectively in the forward view, with an intention to achieve more powerful temporal combinations to achieve enhanced bias-variance trade-offs.
>
> - In [3], multiple Q-learning models are used to predict the return each with a different expected delay. The best-performing model is prioritized. This method is interesting, however, the state space and action space of the application considered in our paper is (probably) too large to handle using q-tables.
>     - The approach appears to be based on a single delay and not a distribution, while our context has a distribution of delays.
>     - Also, their delays appear to be exogenous, that is the delays do not depend on the action or policies, while our case has the delay being endogenous or dependent on the action or policy.
>
> Reviewer:
>
> The authors motivate the predictron submodule by « highly delayed rewards » (Sec. 3.3 and a few other places). How much is it « high »? Although I guess it is stochastic, is it bounded? Another concern is regarding the way time steps are set in the MDP: according to Eq. (8) time is discretized in order for the reward signal to fit with the MDP framework. But what is a typical value of $\Delta_i:= t_i - t_{i-1}$
> ? Would not it affect that of the reward delay, i.e. high $\Delta_i$ implies lower delay value? Similarly, what is a typical value for $h$ in Sec. 3.3.? I guess this should depend on an average delay value. How would a high $h$ value affect the computational time of PDQN (higher $h$ implies long-term inference)?
>
> Author response:
> - We do not fully understand the question, or whether there is some misunderstanding. To respond: First, the delay is bounded, since queuing delays are bounded, in any real factory system. Next, the “highly delayed reward” is contextual rather than absolute, since it concerns the number of other decisions that ultimately also impact the final reward, and whose confounding effects need to be addressed in assigning credit to the current decision. We think that rewards with delays exceeding around 50-100 decision steps could be considered “highly delayed”. In our case, the delays could be several hundred or several thousand time-steps, with a comparable number or more of decisions.
> - To clarify, the delay is determined by the actions taken by the agent and the due time for the parts handled. The longest delays are seen from actions taken immediately after the release of a batch, as these actions are not rewarded until the parts in the batch are completed. The number of actions taken can vary, however, we estimate that to complete 1 batch with 9 head types in a system with 30 batches and 20 sequence steps, the longest delay related to an action can be approximated to 9x30x20=5400 actions in the whole system, till the part acted upon by the action exits the system and thus stops to receive rewards.
>    - However, as actions on some parts also affect the flow of other parts, and as some of the rewards start to appear when the parts start to get overdue, the exact number for the number of time steps a reward is delayed may vary.
> - As the reviewer correctly notes, the $\Delta_i$ is varying as well. It varies from 0 seconds (no time between actions) to several minutes. It depends on the completion time of the next parts to be processed.
>    - This is also one of the reasons for the very high variance of the target.
> - The h-value determines the number of samples to consider as part of the horizon-bound target for the predictron. It therefore only will affect the collection of data in the predictron training phase and not the inference time.
>    - Thereby the h-value determines the bias-variance tradeoff for the target for the predictron, where the variance is created by the samples and the bias is based on the Q-network estimation of the remaining return.

---

> ### Author Response · Authors · 2021-11-20
> **Response to review (2/2)**
>
> Reviewer:
>
> How different are Eq. (11) and (12) from Eq. (5) and (6), respectively? If these are in some sense equivalent, the authors can synthesize their notation and refer to (5) and (6) in Sec. 3.3. instead of rewriting them.
>
> Author response:
> - They differ in the sense that eq 11 and 12 are using the horizon bounded (plus q-network estimated remaining value) version of the return Ep[gi |si ] instead of Ep[g |s ].
>  - This could be written out in text, however, we believe that it is easier to follow the paper if the equations are updated to the versions actually used in the PDQN setting.
>
> Reviewer:
>
> Sec. 2 looks like a Preliminary section providing the problem setting, more than a Related work section. In that regard, a review of delayed RL works is missing. More precisely, high delay in the signal results in some challenging issues: although one could perform embedding and retrieve a non-delayed MDP as in [1], this greatly increases computational complexity. In my opinion, the predictron has a major advantage over augmentation methods that may be even less tractable than MCTS (such drawback of MCTS is indeed mentioned in Sec. 3.3).
>
> Author response:
> - This is a very good point. We strongly agree with the reviewer. Many of the prior research results, on incorporating delays, whether constant or stochastic, result in some form of state augmentation, and a blowing up of the associated state spaces. Furthermore, they appear to be very non-robust when incorporating many of the stochastic delays, particularly when they are action-dependent and endogenous, rather than external/exogenous.
> - We appreciate the inputs, which would help us explain and position the importance of our work better, by explaining the very limited results under very constrained situations available in the literature, with very strong assumptions or approximations. To discuss the papers provided by the reviewer the following points have been added to the paper in section 2:
>    - [1] assumes fixed or deterministic delays or stochastic delays which are independent of the state.
>    - [2] considers MDPs in which there are action delays such that actions are executed some number of steps after they are chosen. This is a separate issue from the one we are addressing with PDQN. In our case actions are executed immediately, the challenge in our case is that actions affect the return over many timesteps, that is, result in highly delayed rewards.
>    - [3] uses a sequence of assumptions and approximations, which may be highly limiting and potentially not applicable, as described in the answer to one of the above questions.
>    - [4] is limited to fixed, constant delays, which is not applicable for our case.
>
> Reviewer:
>
> In [2], the authors use a forward model to tackle execution delay issues. How does the predictron compare to this type of model? The use of eligibility traces seems to have similar effects.
>
> Author response:
> - This paper is addressed along with the others in the previous question and has been added to section 2 in the paper
>
> Reviewer:
>
> In Figs. 2-3, DQN seems to perform decently against PDQN. This discredits a bit the use of the predictron. How could we explain such good performance of DQN which, by the way, does not account for delay at all? Unexpectedly, although they are fashioned precisely to solve such scheduling problems, FIFO and CR perform much worse than those two deep RL methods. How can this be interpreted?
>
> Author response:
> - Although the DQN does perform decently, the PDQN still outperforms it in most systems tested. Our testing showed that the predictron is able to provide improved targets for learning. However, there may be some instability in the learning from the iterative switching between training of the predictron network and the policy network that prevents the policy network from taking full advantage of this. We are working to refine the learning algorithm to improve on this.
> - PDQN incorporates the power of RL, which is a state-dependent approach, therefore we expected it to be able to outperform FIFO and CR as:
>   - FIFO is state-independent, unlike an RL version.
>   - CR is not designed to anticipate future failures and is thus limited in incorporating the system state.

---

### Author Response · Authors · 2021-11-20
**Response to reviewers (1/2)**

Dear Reviewers

Thank you for your careful review of our paper and your insightful comments.

We have attempted to provide satisfactory answers to your concerns. We have also highlighted the main concerns we noted in the reviewers’ comments and will here suggest alternative positioning of the paper. We hope that these significant clarifications will enable the reviewers to revise their evaluation(s) significantly upward.

We begin by addressing the motivation, intuition, and theoretical underpinnings of the PDQN.
1. We first discuss the desired elements of a (model-based/planning) approach to address a system with the degree of complexity, dynamics, and stochastics presented by our manufacturing system.<br>
Our interest and motivation are to develop:
   1. A model-based or planning-based approach, due to the availability of the (manufacturing) system model (via the simulators of the system), to compare it with model-free approaches such as DQN and its variants, to understand the sources of enhanced performance, if any. Note that the literature has had results that indicate both enhancement and degradation of performance when using model-based approaches. In addition, we would like to incorporate effective temporal bias-variance trade-offs.
   2. Finally, we would like a powerful and effective representation via an “effective” state
   3. We would like the approach to be computationally feasible.
   4. We need to explicitly or implicitly incorporate highly delayed/late and sparse rewards, if possible. We need to accommodate policy-dependent delayed rewards in particular. Note that these policy-dependent delays have distributions, rather than having fixed values.

2. Recent methods such as Value Prediction Networks [5] capture the model-based/planning, and representational elements. However, they lack the bias-variance capability of eligibility traces, especially as encapsulated in the lamba return. MuZero [6] is a more powerful version of this planning and representational combination. However, it too lacks the bias-variance capabilities provided by eligibility traces or lamba returns.
3. We now explain below the intuition underlying the development of the PDQN, which incorporates all four of the above desirable features, and follow up with a discussion of associated concepts and theory.

We now clarify some of the rationale and theory underlying the PDQN, as this seemed to be a source of some (or very significant) misunderstanding.
1. The Manufacturing system considered has both high and distributed delays of rewards. Furthermore, it has highly stochastic state transitions, thus making works such as Value Prediction Networks [5] and MuZero [6] inadequate, as they both seek to approximate the state transitions explicitly. The predictron architecture plans in abstract reasoning steps, which only seeks to estimate the value from each reasoning step without the notation of state transitions.
2. A key observation is that the Predictron architecture incorporates planning via the depth feature. This in turn implies that the Predictron-based target incorporates implicit planning, while most other approaches either lack this planning feature, have much cruder planning, or have very computationally intense planning.
3. The Predictron also implements a bias-variance tradeoff by combining abstract reasoning steps (the depth) with learned eligibility trace parameters in a forward view manner.
4. This way, the PDQN achieves planning as part of the value prediction represented as the target.
5. The PDQN gains 4 powerful elements through the forward lambda return in the predictron:
   1. Implicit planning
   2. Temporal bias-variance enhancement
   3. Appropriate value-based representation as opposed to state-based representations of future predictions.
   4. In addition, the PDQN also addresses delayed rewards through offline planning, where the episodes generated in the simulations pick up the effect of the “very long-delayed” rewards.

[5] Junhyuk Oh, Satinder Singh, and Honglak Lee. Value prediction network. In I. Guyon, U. V. Luxburg, S. Bengio, H. Wallach, R. Fergus, S. Vishwanathan, and R. Garnett (eds.), Advances in Neural Information Processing Systems 30, pp. 6118–6128. Curran Associates, Inc., 2017.

[6] Julian Schrittwieser, Ioannis Antonoglou, Thomas Hubert, Karen Simonyan, Laurent Sifre, Simon Schmitt, Arthur Guez, Edward Lockhart, Demis Hassabis, Thore Graepel, Timothy Lillicrap, and David Silver. Mastering atari, go, chess and shogi by planning with a learned model. Nature, 588 (7839):604–609, December 2020.

---

### Author Response · Authors · 2021-11-20
**Response to reviewers (2/2)**

We now address some of the general concerns.
1. Benchmarks using planning
   1. The reviewers suggest using a more recent method as a benchmark.
   2. Our preliminary work did not show improved performance from using the different DQN extensions described below. We, therefore, did not include these methods, as they seemed unsuited to the task.
   3. It would be interesting to use other planning methods as benchmarks
      - However, in our preliminary work, we tested MuZero, which did not converge on the problem.
         - The associated computational load appeared to be too heavy.
      - Extensions to DQN: The extensions to the DQN we tried include N-step DQN, Double Dueling DQN, Prioritized Experience Replay, and the full Rainbow setup. To repeat, no significant performance gains were achieved, when compared with the base DQN.
         - We expect that the basic DQN, and its extensions, are not able to utilize the information from longer horizons, as the variance of the return is very high.
         - The bias from the Q-network resulting from the use of the Predictron target, therefore, appears to be the main driver for the performance achieved by the DQN.

2. Our objective function versus other alternatives
   1. The objective we consider is important in this domain. This objective function (in isolation), or variations, has been studied for several decades, without closure in terms of optimality. We agree, however, that additional components could be added to the objective to increase the practical usefulness. Since we are introducing a new algorithm in this paper, we chose to limit the scope of the problem the method is solving.
   2. Future work will consider expansions of the objective to consider additional productivity indicators. Consequently, the paper has been updated to emphasize that the Additional Observations section is meant to serve as the motivation for future research. We could also move it to the Appendix, to avoid any possible confusion, since only Section 4.3: MAIN RESULTS, contains results relevant to the objective function that is working with, in this paper.

3. Relevant literature
   1. The relevant literature section has been expanded to address additional methods which account for delays
   2. Many of the other methods are not applicable to this problem as they either address different aspects of delay or are modeled in a manner that is incompatible with our problem formulation, such as constant or stochastic delays, which do not account for policy dependent delay distributions. Admittedly, we do not address or model the delay directly, since that problem has been shown to be very difficult in previous research, with an explosion of the state space implying computation infeasibility, with limited performance gains. Thus, our approach is more intuitive and more heuristic, and less formal, though very clearly motivated by theoretical structures and analyses. We note that papers [5], [6} below, on V, and MuZero, are similarly heuristic, though well motivated.

We would be happy to incorporate any changes incorporating one or more of these responses into the revised version of the paper if the reviewers consider it desirable.

[5] Junhyuk Oh, Satinder Singh, and Honglak Lee. Value prediction network. In I. Guyon, U. V. Luxburg, S. Bengio, H. Wallach, R. Fergus, S. Vishwanathan, and R. Garnett (eds.), Advances in Neural Information Processing Systems 30, pp. 6118–6128. Curran Associates, Inc., 2017.

[6] Julian Schrittwieser, Ioannis Antonoglou, Thomas Hubert, Karen Simonyan, Laurent Sifre, Simon Schmitt, Arthur Guez, Edward Lockhart, Demis Hassabis, Thore Graepel, Timothy Lillicrap, and David Silver. Mastering atari, go, chess and shogi by planning with a learned model. Nature, 588 (7839):604–609, December 2020.

---

### Decision · Program_Chairs · 2022-01-20

**Decision:**

Reject

**Comment:**

This paper deals with solving the problem of scheduling machines in a semiconductor factory using an RL approach. As the different actions take a different amount of time to complete, the authors propose to use a predictron architecture to estimate the targets in DQN. The experimental results show that the proposed method outperforms the considered baselines on two scheduling problems.

After reading the authors' feedback and discussing their concerns, the reviewers agree that this paper is still not ready for publication.
In particular, the main issues are about the novelty/similarity with respect to related works, the lack of theoretical insights and formal definitions, the effectiveness of the presented benchmarking, lack of analysis of some unexpected results.

I encourage the authors to take into consideration the concerns raised by the reviewers when they will work on the updated version of their paper.